# 4D-VQ-GAN: A World Model for Synthesizing Medical Scans at Any Time Point for Personalized Disease Progression Modeling of Idiopathic Pulmonary Fibrosis

**An Zhao**[*1,2,3] (ID)                                                AN.ZHAO.19@UCL.AC.UK

[1] *Hawkes Institute, University College London*

[2] *Department of Computer Science, University College London*

[3] *Satsuma Lab, Hawkes Institute, University College London*

**Moucheng Xu**[*†1,3,4]                                        MOUCHENG.XU.18@UCL.AC.UK

[4] *Department of Medical Physics and Biomedical Engineering, University College London*

**Ahmed H. Shahin**[1,2,3]                                      AHMED.SHAHIN.19@UCL.AC.UK

**Wim Wuyts**[5]                                                 WIM.WUYTS@UZLEUVEN.BE

[5] *Department of Respiratory Medicine, University Hospitals Leuven*

**Mark G. Jones**[6]                                             MARK.JONES@SOTON.AC.UK

[6] *NIHR Southampton Biomedical Research Centre and Clinical and Experimental Sciences, University of Southampton*

**Joseph Jacob**[‡1,3,7]                                         J.JACOB@UCL.AC.UK

[7] *UCL Respiratory, University College London*

**Daniel C. Alexander**[‡1,2]                                    D.ALEXANDER@UCL.AC.UK

**Editors:** Accepted for publication at MIDL 2025

## Abstract

Understanding the progression trajectories of diseases is crucial for early diagnosis and effective treatment planning. This is especially vital for life-threatening conditions such as Idiopathic Pulmonary Fibrosis (IPF), a chronic, progressive lung disease with a prognosis comparable to many cancers. Computed tomography (CT) imaging has been established as a reliable diagnostic tool for IPF. Accurately predicting future CT scans of early-stage IPF patients can aid in developing better treatment strategies, thereby improving survival outcomes.

As inspired by the recent success of world models in generating video-based virtual physical worlds, we present the first world model for IPF, to synthesize realistic scans of early-stage IPF patients at any time point. We term our model 4D Vector Quantised Generative Adversarial Networks (4D-VQ-GAN). Our model is trained using a two-stage approach. In the first stage, a 3D-VQ-GAN is trained to reconstruct CT volumes. In the second stage, a Neural Ordinary Differential Equation (ODE) model is trained to capture the temporal dynamics of the quantised embeddings, which are generated by the encoder trained in the first stage. For clinical validation, we conduct survival analysis using imaging biomarkers derived from generated CT scans and achieve a C-index either better than or comparable to that of biomarkers derived from the real CT scans. The survival analysis results suggest the potential clinical utility inherent to generated longitudinal CT scans, showing that they can reliably predict survival outcomes. The code is publicly available at https://github.com/anzhao920/4DVQGAN.

**Keywords:** 4D image synthesis, world models, VQ-GAN, neural ODEs, spatial-temporal disease progression modelling, CT, IPF

---

* Contributed equally

† Work completed at UCL, Moucheng has now moved to Medtronic.

‡ Joint senior authors

## 1. Introduction

Understanding the disease progression trajectories is vital for understanding diseases' biological mechanisms and developing better treatments. The traditional methods for understanding the disease progression focus on modelling the progression trajectories using longitudinal, low-dimensional data (e.g., differential equation models (Villemagne et al., 2013; Jack et al., 2013; Oxtoby et al., 2014), self-modelling regression methods (Jedynak et al., 2012; Donohue et al., 2014), event-based models (Wijeratne and Alexander, 2024; Young et al., 2018; Fonteijn et al., 2012)). Those traditional methods normally require strong assumptions of the statistical disease progression patterns, relying on the prior knowledge of the diseases, which are hard to obtain for rare diseases. In addition, the majority of the traditional methods models the progression trajectories at a population level, hindering their applicability on hetergeneous diseases.

To address the aforementioned limitations of the traditional methods to understand the disease progression, we explore a new paradigm of machine learning methods, namely, world models (NVIDIA et al., 2025; Parker-Holder et al., 2024; Bruce et al., 2024). The world models simulate the physical worlds with interactive virtual environments. For example, a recent world model called Genie (Parker-Holder et al., 2024; Bruce et al., 2024) can create game video based environments for the AI agents to interact with. Similarly, in the context of the disease progression, we can adapt temporal medical imaging as the environments, on the analogy as using the videos as environments in physical world models.

In this paper, we explore world models on investigating the progression trajectory of idiopathic pulmonary fibrosis (IPF), a progressive lung condition, with high-resolution CT volumetric scans. As proof of the concept, in this paper, we focus on only one interactive action with the virtual progression trajectories, which is the time. More precisely, we introduce the 4D Vector Quantized Generative Adversarial Network (4D-VQ-GAN). Given two 3D CT scans of an IPF patient at irregular time points, our method can generate synthetic 3D images at any desired time point, effectively modelling a virtual continuous disease progression trajectory for each individual. More importantly, we found that biomarkers derived from the generated CT volumes exhibit a strong clinical correlation with survival outcome, highlighting the potential of our method for personalized treatment planning.

## 2. Related Works

Our work can be regarded as a type of spatial-temporal disease progression modelling with high-dimensional volumetric imaging data. Previously, a few generative models have been explored to simulate longitudinal magnetic resonance imaging (MRI) scans specifically in the context of Alzheimer's disease (AD) progression (Couronné et al., 2021; Sauty and Durrleman, 2022; Kim et al., 2021; Martí-Juan et al., 2023; Fan et al., 2022; Puglisi et al., 2024; Ravi et al., 2022; Yoon et al., 2023). For instance, auto-encoder styles model have been used to couple with linear temporal models in (Sauty and Durrleman, 2022; Kim et al., 2021). However, they rely on oversimplified assumptions about evolution trajectories (e.g. linear progression). Other methods (Martí-Juan et al., 2023; Fan et al., 2022) use recurrent neural networks to capture the temporal information. These methods model dynamics in discrete steps, limiting their ability to capture the continuous nature of disease progression. Puglisi et al.(Puglisi et al., 2024) use the latent diffusion model and incorporate prior

knowledge to model the disease progression. However, it also struggles to ensure continuous temporal changes. In addition, existing methods for generating longitudinal MRI scans in AD are not directly applicable to CT lung scans of IPF patients for several reasons. First, IPF is much rarer with shorter patient lifespans than AD resulting in less imaging being acquired. Furthermore, radiation side effects from CT scans discourage repeated scans, resulting in a scarcity of longitudinal CT data and hindering model development. Second, lung CT scans contain vastly more fine textured-structures like vessels, airways, and interstitial tissue compared to the smooth, homogenous brain seen in MRI. Generating these intricate lung structures synthetically is more challenging than replicating brain tissue.

## 3. Methods

Our 4D-VQ-GAN is a self-supervised generative model trained on temporal CT volumes. As shown in Figure 1, the model consists of two key components: a 3D-VQ-GAN and a temporal model. We train the two components of our model sequentially to make the overall training on high-dimensional spatial-temporal data easier to stabilize. In the first stage, we train a 3D-VQ-GAN to reconstruct each scan. By doing so, the trained encoder of the 3D-VQ-GAN can project the high-dimensional data into low-dimensional code books. In the second stage, we use a Neural ODE to capture the temporal dynamics of the code books. Most importantly, the neural ODE provides us an interactive environment with the latent temporal disease trajectories, so that during the inference, we can simulate the scans at any time point given two scans. In addition, the use of code books as input for temporal modelling avoids overfitting on the temporal data.

**3D-VQ-GAN** As shown in Figure 1, the first stage of training involves a 3D-VQ-GAN (Ge et al., 2022) to reconstruct CT volumes for each case at every time point in the training set. Unlike the original 2D-VQ-GAN (Esser et al., 2021), our 3D-VQ-GAN employs 3D convolutional layers, enabling it to capture spatial structures more effectively in volumetric imaging data. Following the VQ-VAE framework (Van Den Oord et al., 2017), 3D-VQ-GAN compresses high-dimensional volumetric imaging data into a discrete set of latent codes, constrained by a predefined codebook size. Since the model is trained on imaging data, its codebook learns to represent meaningful imaging patterns, with each code acting as a compact and discrete representation of local anatomical structures or texture features. We adapted the original 3D-VQ-GAN loss functions (Ge et al., 2022) for training. Further details can be found in Appendix E.2.

**Temporal Model** In the second stage of the training, we train a temporal model that can reconstruct the temporal trajectories of the latent embeddings ($z$) of the imaging data from different time-points, as demonstrated in Figure 1. We realise that generating the future or past latent embeddings from a few observed latent embeddings naturally formulates as an ordinary differential equation. We therefore utilise a neural ODE solver (Chen et al., 2018) to predict the unknown embeddings at new time-points. We found that it is more beneficial to adapt a 3D-ConvGRU (Ballas et al., 2015) as the encoder of the neural ODE solver on our data, with much better computational efficiency. The outputs of the neural ODE are then fed into a light-weight projector, two 3D convolutional layers, to reconstruct the latent embeddings $z$ at inquired time points, with a specified time interval. In practice, We found that adding skip connections to the generated embeddings improves results,

Stage 1: Train the 3D-VQ-GAN to reconstruct the CT volumes

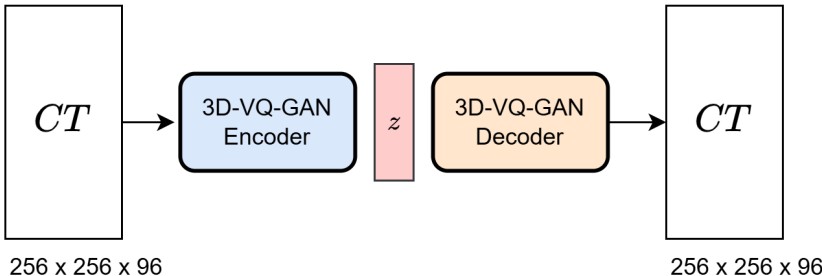

Stage 2: Train the temporal model to reconstruct the temporal sequence of $z$ from stage 1

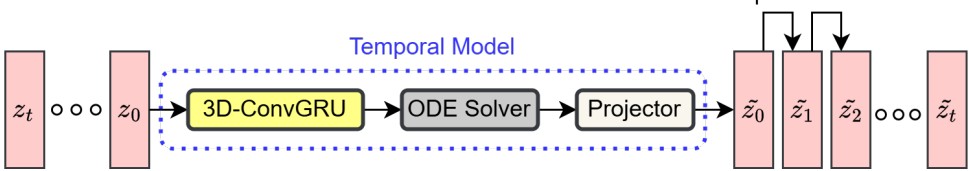

Figure 1: The overview of our two-stage training strategies. The first stage trains an encoder-decoder-based 3D-VQ-GAN to reconstruct the CT volumes. The second stage takes the latent embeddings $(z_t, ..., z_0)$ from the first stage, and trains a temporal model to reconstruct them. The temporal model consists of a 3D-ConvGRU, that compresses the temporal latent embeddings to match the dimensionality of the input of the ODE Solver to ease the computational burden. The projector, a light-weight 3D convolutional module reconstructs the temporal latent embeddings from the outputs of the ODE Solver. Those reconstructed latent embeddings are then fed into a frozen 3D-VQ-GAN Decoder from stage 1 for longitudinal CT reconstruction.

offering two key benefits. 1) Training Stability and Efficiency: Skip connections stabilize training and speed up convergence, similar to ResNets, by focusing gradient propagation on meaningful pathological changes rather than both stable anatomy and disease progression. 2) Anatomical Preservation: Skip connections condition predictions on baseline anatomy, preserving stable structures and enabling personalized disease progression modelling. We train the temporal model in a self-supervised manner by using a L2 loss between the input embeddings and the reconstructed embeddings. Please refer to Appendix E.3 for details.

**Inference** The inference follows the process shown in Figure 2. The trained model needs two CT scans. The user can also input two hyper-parameters for the neural ODE, namely the interval time and the total time duration. For example, as shown in Figure 2, given two initial scans at time point 0 and 1, the interval as 1 year and the total time duration as 3 years, the model will reconstruct the scans at time point 0 and 1, and start to extrapolate future scans at time point 2 and 3.

**Survival analysis and biomarker discovery** We explore our model's clinical utility using a survival analysis based approach to mimic the clinical workflows. Radiologists

track prognostic imaging biomarkers in IPF over time to assess disease progression. Though we lack comprehensive visual scores for all cases, we propose a method that mirrors clinical workflows, including selecting key prognostic biomarkers, analyzing their longitudinal changes, and comparing their prognostic value in synthesized vs. real scans. Despite not being pixel-perfect replicas, if synthesized scans maintain consistent biomarker representation and exhibit comparable longitudinal changes to real scans, it would support their clinical utility. The method is as follows:

- Biomarker Extraction: The trained 3D-VQGAN encoder is used to project each CT scan into a set of codebook. Each code index represents a distinct imaging pattern in the CT. The frequency of each code index reflects the prevalence of the corresponding imaging pattern. We then use these normalized frequencies of code indices as candidate cross-sectional prognostic biomarkers.

- Cross-sectional prognostic biomarker selection: To identify prognostic cross-sectional biomarkers, we perform survival analysis using five-fold cross-validation. The original training dataset is divided into training and validation datasets. In each fold, univariate Cox proportional hazards models (Cox, 1972) are applied to each biomarker, adjusting for age, sex, and smoking status (smoking vs non-smoking). Biomarkers are selected based on three criteria: (1) Hazard Ratio (HR) $> 1$ (indicating that higher biomarker values are associated with increased mortality risk), (2) p-value $<$ 0.01 (ensuring statistical significance), and (3) Concordance Index (C-index) $> 0.5$ (demonstrating predictive ability). A biomarker must meet these criteria in all five folds to be considered robust. The top five biomarkers with the highest mean C-index values across the validation sets are selected for further longitudinal biomarker analysis, ensuring that the most predictive biomarkers are prioritized.

- Longitudinal biomarker analysis: For each patient in the test dataset, we use their first two available CT scans to synthesize both their second available scan and an additional follow-up scan one year later. The synthetic second scan serves as a predicted version of the actual second scan. We then extract the five selected cross-sectional biomarkers from both real and synthetic scans at two time points: the second scan and the one-year follow-up. The longitudinal biomarker is defined as the change in these biomarkers between the two time points, separately computed for real and synthetic scans. These longitudinal biomarkers, along with the covariates, are input into the Cox model to assess their prognostic value in the test dataset. This analysis evaluates the consistency of biomarker trajectories between real and synthetic scans, and explores the potential utility of synthetic scans in tracking changes over time.

## 4. Experiments

**Datasets** Our data comes from a longitudinal dataset comprising 681 volumetric CT scans from 219 IPF patients, obtained from University Hospitals Leuven, Belgium, a single centre in Leuven. We randomly divided the dataset into 80% for training (552 CT scans from 175 patients) and 20% for testing (129 CT scans from 44 patients). Different patients have varying numbers of scans, reflecting the natural variations in imaging availability in real

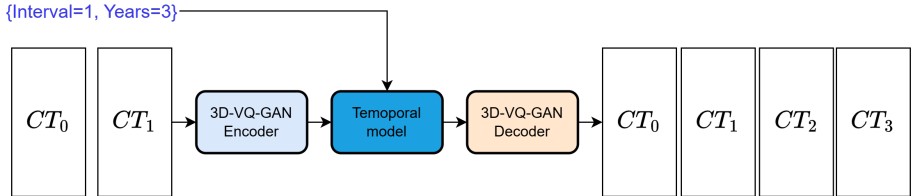

Figure 2: The inference of the trained model. Given two scans, the model can generate more scans up to the specified year at a specific interval.

clinical practice. The CT scans were acquired with varying slice thicknesses, ranging from 0.75 mm to 1 mm, and in-plane resolution varied with pixel spacing ranging from 0.38 mm to 0.98 mm. Additionally, we use an external cross-sectional dataset of 98 IPF patients from the University Hospital Southampton NHS Foundation Trust, UK, to evaluate the generalizability of the 3D-VQ-GAN trained in Stage 1 and more details can be found in Appendix D.1. For survival analysis, patients must have complete data on death status, censoring information, and all relevant covariates (age, sex and smoking status). In the training dataset, 71 out of 141 patients have observed deaths. For the cross-sectional biomarker analysis test dataset, patients must have at least three time points, with 10 out of 17 patients having observed deaths. For the longitudinal biomarker analysis test dataset, patients are required to have at least three time points and a one-year follow-up for the second available data point, with 9 out of 14 patients having observed deaths.

**Preprocessing** We focus on modelling changes within the lung areas. We segment the lung regions using a pre-trained U-Net (Ronneberger et al., 2015; Hofmanninger et al., 2020) for all CT scans of IPF patients and visually inspect the lung masks. Subsequently, we register the longitudinal lung scans to remove extraneous artifacts caused by lung motion or incorrect body positioning. Our lung scan registration method is a faster version (implemented in (Hansen and Heinrich, 2021)) of the CorrField method (Heinrich et al., 2015). Visualizations of the segmentation and registration outcomes can be found in Appendix C.

**Training** Our models are trained on an NVIDIA A100 80GB GPU. The first training stage used a batch size of 1, with an accumulated batch size of 6, and lasted 20,000 steps, equivalent to 10 days. The second training stage also used a batch size of 1 and lasted 15 hours. The hyperparameters of the training can be found in Appendix A.

**Evaluation Metrics** To evaluate the image quality of the reconstructed CT scans in the first stage of training, we use Mean Squared Error (MSE). For assessing the image quality of the generated temporal CT scans in the second stage, we use Peak Signal-to-Noise Ratio (PSNR) and Structural Similarity Index (SSIM). For survival analysis, we use C-index, a metric that measures the predictive accuracy of a model by evaluating the agreement between predicted risk scores and actual survival outcomes.

## 5. Results

In this section, we present the results of our model on the following tasks: 1) interpolation, for imputing missing CT scans between the two input scans; 2) extrapolation, for predicting

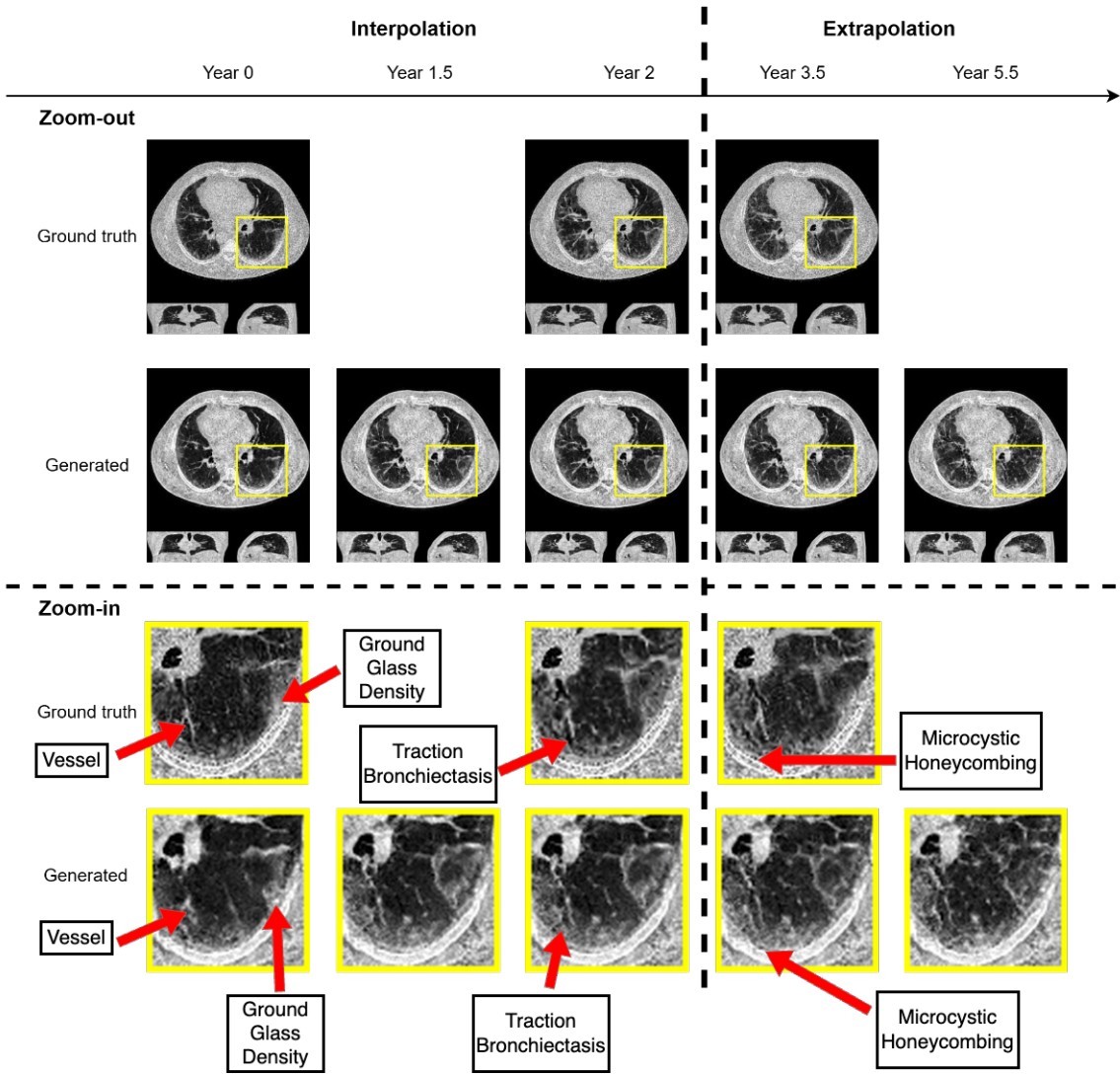

Figure 3: Three real CT scans of an IPF patient are shown in the upper panel, representing axial, coronal, and sagittal sections. Using two scans from year 0 and year 2, the trained model can generate CT scans at any arbitrary time points. The below panel shows the generated CT images at five different time points, with three corresponding to the real scans. A zoomed region of the left lower lobe (yellow box) in the real and generated CT scans show comparable amounts of architectural distortion, patterned ground glass opacification and reticulation, all hallmarks of lung fibrosis. The availability of our scans are not uniform across time and across patients, the model is trained on scans at irregular time points.

the future CT scans beyond the time span of the two given input scans; 3) survival outcome prediction, for evaluating of the clinical utility of the proposed method.

Table 1: Ablation Study on Interpolation and Extrapolation Tasks.

| Model Variant | Interpolation | | | Extrapolation | | |
|---|---|---|---|---|---|---|
| | MSE↓ | SSIM↑ | PSNR↑ | MSE↓ | SSIM↑ | PSNR↑ |
| 3D-VQ-GAN+ODE encoder (ConvGRU) | 0.018 | 0.500 | 17.913 | 0.023 | 0.472 | 16.974 |
| 3D-VQ-GAN+ODE encoder (ConvGRU)+embeddings skip connections (proposed) | 0.020 | 0.469 | 17.387 | 0.019 | 0.489 | 17.816 |
| 3D-VQ-GAN+ODE encoder (ODE-ConvGRU)+embeddings skip connections | 0.018 | 0.508 | 17.900 | 0.026 | 0.440 | 16.913 |
| 3D-VQ-GAN+ODE encoder (ConvGRU)+embeddings skip connections+masked out inputs | 0.035 | 0.343 | 15.366 | 0.036 | 0.340 | 15.24 |

**Interpolation and extrapolation** In this experiment, we evaluate the performance of the proposed method in reconstructing latent dynamics from sparse, irregularly sampled data. The model is trained to reconstruct the given input sequence of 3D imaging data and is then tested on both interpolation and extrapolation tasks. Figure 3 illustrates example qualitative results. The patient's year 0 and year 2 scans were used for inference, with the year 3.5 scan serving as ground truth for the extrapolation experiment. Inference was performed with a 0.5-year interval, extending up to 5.5 years from the baseline (year 0). As shown in the zoomed-in section, both real and generated CT scans exhibit comparable architectural distortion, ground glass opacification, and reticulation—key features of lung fibrosis. The quantitative results are shown in Table 1.

**Ablation study** The proposed method uses Neural ODE to predict the difference embeddings for each time point relative to the previous one, incorporating skip connections. In the ablation experiments (Table 1), we remove the skip connections and allow the model to directly predict the latent embeddings of all time points using Neural ODE. However, this modification resulted in significantly poorer extrapolation performance. We also replaced ConvGRU with ODE-ConvGRU (Park et al., 2021) as the ODE solver encoder. While ODE-ConvGRU is designed for irregularly sampled longitudinal data, it leads to notably worse extrapolation performance compared to the model using standard ConvGRU. Additionally, we experiment with randomly masking time points (excluding the baseline) and task the model with reconstructing the unseen time points. This approach also results in significantly worse extrapolation performance. Overall, the proposed model demonstrates balanced performance in both interpolation and extrapolation tasks.

**Survival outcome prediction** Following the method in Section 3, the top five most significant cross-sectional imaging biomarkers are selected for survival analysis. The selected five biomarkers exhibit weak to moderate correlation, with pairwise correlation coefficients all below 0.64 and an average correlation of 0.41. In the test dataset, we extrapolate the third CT scan using the first two available CT scans. For the cross-sectional imaging biomarker, the C-index using the generated third scans is 0.943. In comparison, using biomarkers derived from real CT scans for survival prediction yields a slightly lower C-index of 0.914. Next, we compute the longitudinal biomarker by evaluating the change in these top five significant biomarkers over the course of one year, both for real and generated CT scans. By inputting these longitudinal biomarkers along with covariates into the Cox model, we obtain C-indices of 1.0 for both real and generated CT scans. However, these results may be overestimated due to the limited sample size. The C-index of survival analysis using generated scans is comparable to that of real scans, further highlighting the strong clinical potential of our method.

Table 2: Survival outcome prediction results on biomarkers selected from 5-fold cross-validation.

| Biomarker type | C-Index | |
|---|---|---|
| | Real CT scans | Generated CT scans (ours) |
| Cross-sectional biomarker | 0.914 | 0.943 |
| Longitudinal biomarkers | 1 | 1 |

## 6. Discussion

We aim to understand disease progression by interacting with a virtual disease progression trajectory, which is physically visualized through a medical imaging modality that both clinicians and patients can intuitively perceive. This is a key aspect that distinguishes our approach from previous methods in disease progression modeling, which build mathematical models to explicitly describe disease progression based on a limited set of variables derived from medical images. Our approach enables clinicians to perform direct virtual diagnoses on the generated future scans, thereby avoiding information loss that occurs during the extraction of variables from the scans. Our model is also designed to be scalable, allowing future work to explore larger-scale implementations, with more potential interactions (e.g interventions) with the progression trajectories. We hope that our work will interest the community in further exploration of disease progression trajectories through the lens of world models, extending beyond IPF. This approach could as well potentially improve existing screening and prevention technologies for different diseases.

Beyond disease progression analysis, the model has the potential for applications like data augmentation. However, the proposed method exhibits certain limitations. Firstly, the current approach utilizes a deterministic Neural ODE in the latent space, which assumes a shared disease dynamic across all patients. This might not be ideal for heterogeneous diseases with diverse subtypes and distinct progression patterns. Secondly, a crucial limitation of the current validation process is its reliance on a single dataset for evaluating the model. Finally, the study relies on a single training/validation split due to limited training resource, preventing an assessment of performance variability.

## 7. Conclusion

This paper presents a generative environment to simulate the progression trajectories of IPF patients based on CT imaging. The proposed generative environment can be seen as a world model. Our world model for IPF can realistically synthesize CT images of IPF patients that contain visual hallmarks of lung fibrosis comparable to those in real scans. Most importantly, the generated CT images can yield survival outcomes that are sometimes even more accurate than those derived from the real images.

## Acknowledgments

AZ was supported by CSC-UCL Joint Research Scholarship. MCX was supported by GSK, United Kingdom (BIDS3000034123) and UCL Engineering Dean's Prize. AHS was supported by the Open Source Imaging Consortium (OSIC) https://www.osicild.org. DCA is supported by UK EPSRC grants M020533, R006032, R014019, V034537, Wellcome Trust, United Kingdom UNS113739, Wellcome Trust 221915/Z/20/Z. JJ was supported by the Wellcome Trust 209553/Z/17/Z and 227835/Z/23/Z, the Chan Zuckerberg Initative CZIF2024-009938. JJ is Radiology co-lead for the Open Source Imaging Consortium (OSIC). DCA, and JJ are supported by the NIHR UCLH Biomedical Research Centre, UK. This research was funded in whole or in part by the Wellcome Trust [209553/Z/17/Z and 227835/Z/23/Z]. For the purpose to support future research, the author has applied a CC BY-NC 4.0 copyright licence to any author accepted manuscript version arising from this submission.

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

## Appendix A. Training and implementation details

We train the model for two stages, 3D-VQ-GAN and latent disease ODE. All models are trained on a NVIDIA A100 80GB GPU.

### A.1. 3D-VQ-GAN

In line with recommendations from (Ge et al., 2022; Esser et al., 2021), the training of 3D-VQ-GAN begins on all CT scans in the training set using the reconstruction loss. Subsequently, the GAN loss is introduced after 10,000 steps. Hyperparameters are set as follows: $\lambda_{\text{perc}} = \lambda_{\text{rec}} = 4$ and $\lambda_{\text{GAN}} = 1$. The Adam optimizer (Kingma and Ba, 2014) is employed with a learning rate of $3 \times 10^{-4}$ and $\beta_1 = 0.5, \beta_2 = 0.9$. Training the 3D-VQ-GAN spans 20,000 steps, and the best model, determined by the smallest training loss after adding the GAN loss, is selected. The batch size is set at 1, and the accumulated batch size is 6. Training the first-stage model takes approximately 10 days.

### A.2. Disease ODE

After completing the training of 3D-VQ-GAN, we proceed to train the latent disease ODE using the AdamW optimizer (Loshchilov and Hutter, 2017). The training spans 100 epochs, employing a batch size of 1, a learning rate of $2 \times 10^{-4}$, and $\beta_1 = 0.5, \beta_2 = 0.9$. To enhance the model's performance at later time points, we implement a linearly increasing weights strategy, assigning higher loss weights for later time points. The best model, identified by the smallest training loss, is selected. Training the latent disease ODE takes approximately 15 hours.

### A.3. Implementation details

For the implementation of 3D-VQ-GAN, we adopt a similar network structure as outlined in (Ge et al., 2022). The codebook size is set to $M = 256$ with an embedding size of $c = 16$. We employ a compression rate $r = 4$, calculated as the ratio between $D, H, W$ and $d, h, w$. All input 3D CT scans are resized to $D = 96, H = 256, W = 256$ before being fed into the model. In the latent disease ODE, the neural ODE solver is implemented by stacking three 3D convolution layers. The code is implemented using PyTorch 1.8.

## Appendix B. Pseudo code for the algorithm for biomarker extraction and selection

**Input:** Baseline CT scans, time-to-death, censoring status, age, gender, smoking status
**Output:** A ranked list of prognostically significant biomarkers, with corresponding HR, p-value, and C-index values.

**Feature Extraction:**
Use a trained 3D-VQGAN encoder to extract codebook indices from each CT scan.
Compute a normalized histogram of these indices to obtain proportional representations of imaging patterns (biomarkers).

**Biomarker selection via Five-Fold Cross-Validation:**
**for** *fold* **do**
 Split the dataset into training (80%) and validation (20%) sets.
 **for** *biomarker candidate* **do**
  Fit a univariate Cox proportional hazards model, adjusting for covariates (age, sex, smoking status).
  Compute p-value, hazard ratio (HR) from the trained Cox model, and compute C-index on the validation set.
  Retain biomarkers that satisfy $HR > 1$, p-value $< 0.01$, and C-index $> 0.5$ on all five folds.

Among the shortlisted biomarkers, select the top 5 with the highest mean validation-set C-index across folds.

## Appendix C. Pre-processing

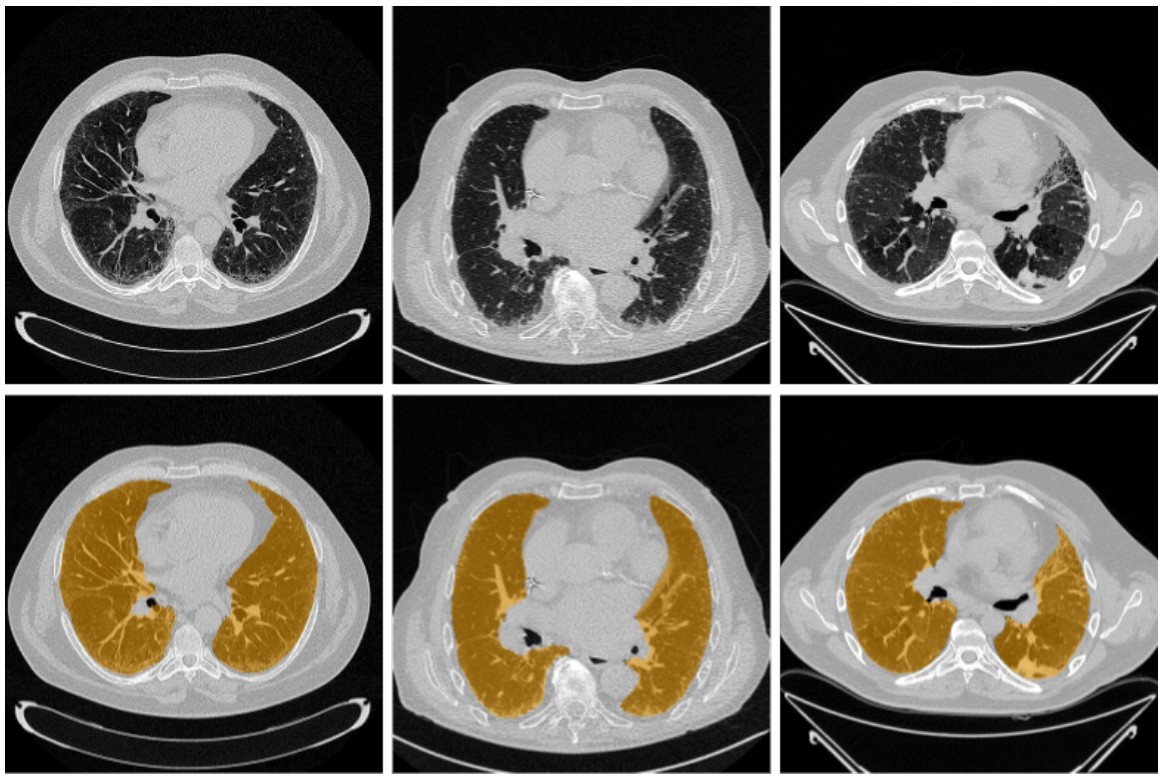

Figure 4: Segmentation results for selected cases from Leuven cohort

The registration process aligns corresponding structures across scans, ensuring the model focuses on disease-related changes. The Learn2Reg challenge, associated with MICCAI 2020 and 2021, provides valuable insights for choosing a suitable registration method for lung CT scans. This challenge compared various approaches on clinically relevant tasks. In the lung CT task, the goal was to register expiration scans to inspiration scans, with corresponding landmarks provided for accuracy evaluation (Hering et al., 2022). Notably, CorrField (Heinrich et al., 2015), a non-rigid registration method, emerged as the top performer among 15 methods, including deep learning and conventional approaches. CorrField achieved a target registration error (the Euclidean distance between corresponding landmarks in the warped fixed and moving scan) of only 1.75mm (Hering et al., 2022).

We use the faster version (implemented in (Hansen and Heinrich, 2021)) of CorrField method (Heinrich et al., 2015), a non-learning-based unsupervised method. CorrField first employs Foerstner operator (Förstner and Gülch, 1987) to extract distinctive keypoints in one 3D volume. Then a dissimilarity distribution over a densely quantized space of displacements is calculated. Finally, a parts-based model is used to infer the smooth motion of connected keypoints and regularize the correspondence field. Specifically, a minimum spanning tree (MST) is generated from the set of sparse keypoints, which enables exact message passing using belief propagation on the graph to regularise the displacement costs.

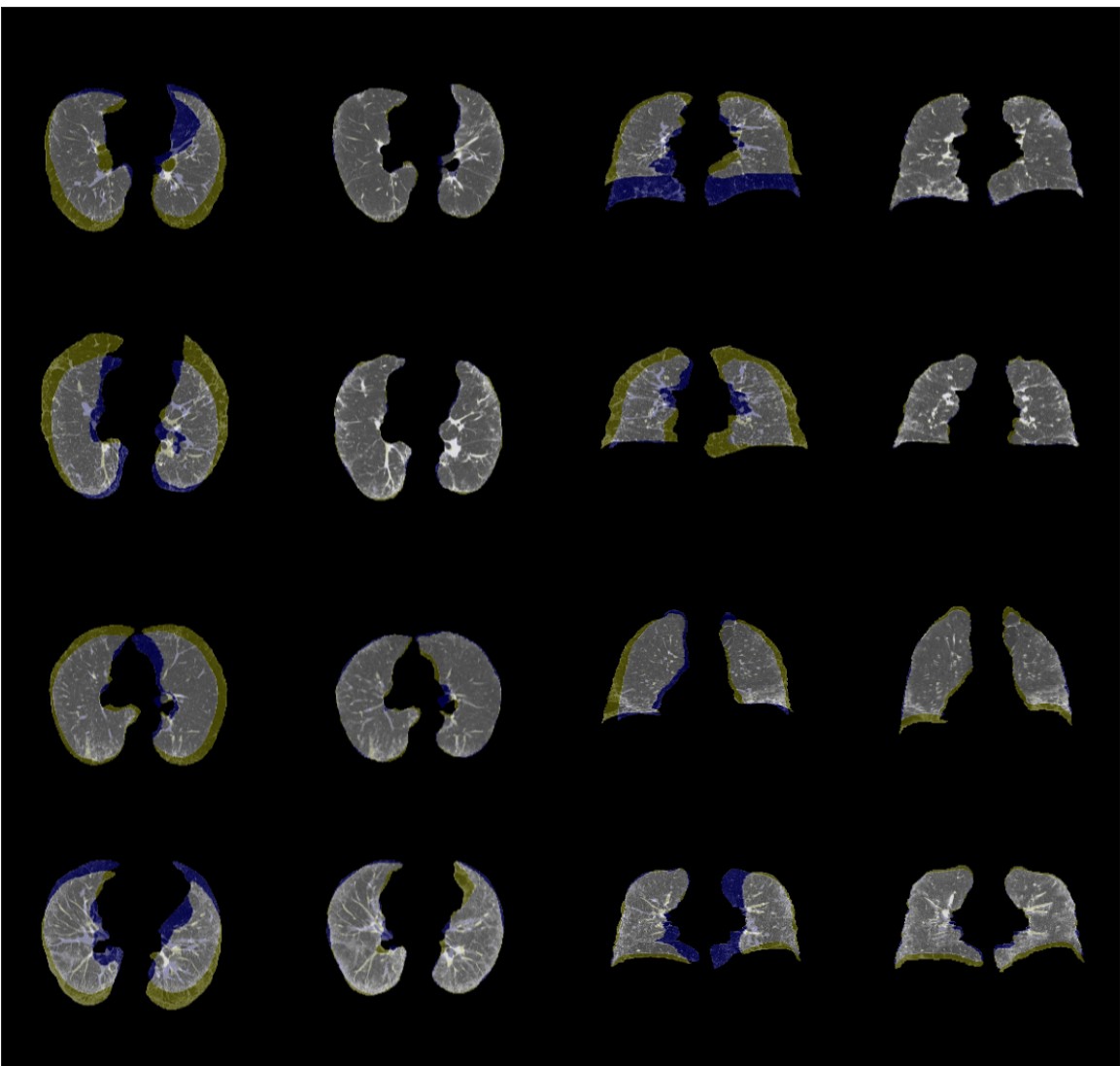

Figure 5: Visualization of four registration outcomes with a focus on lung areas for clarity. The left two columns present axial views before and after registration, while the right two columns showcase coronal views. Baseline scans are denoted in blue, whereas follow-up scans are highlighted in yellow. The merging of colours results in grey or white hues, indicating aligned structures due to RGB amalgamation. Notably, follow-up scans are registered to their corresponding baseline scans. The first two rows illustrate cases with successful registration outcomes, while the subsequent two rows demonstrate instances of varying degrees of misalignment.

To ensure the disease progression model focuses solely on lung tissue, we define keypoints within lung masks for registration. After aligning the longitudinal CT scans to the baseline

CT scan using image registration, we replaced the non-lung region in the registered scans with the corresponding region from the baseline CT scan. This eliminates distractions from surrounding body parts. We leverage the default hyperparameters from the validated implementation (https://grand-challenge.org/algorithms/corrfield/) for registration. To verify registration quality, all registered scans undergo visual inspection by me to identify and exclude those with significant errors. However, because of the non-clinical background, there is still risk compared with verification conducted by experienced radiologists.

## Appendix D. Additional experiments

In this section, we began by conducting experiments on the initial stage of the model to show the reconstruction performance of 3D-VQ-GAN, both qualitatively and quantitatively. Additionally, we explore the impact of varying hyperparameters on the reconstruction performance. Subsequently, we assess the proposed two-stage model's effectiveness in disease progression modelling and interpolation tasks, demonstrating its capability to capture the dynamics of disease progression. Finally, we present visualizations of the learned codebook for enhanced interpretability.

Table 3: Reconstruction performance of 3D-VQ-GAN with different hyperparameters.

| No. | Vocabulary size | Compression rate | Learning rate | MSE on internal test set | MSE on external test set |
|---|---|---|---|---|---|
| 1 | 1024 | 4 | $3 \times 10^{-4}$ | $3.76 \times 10^{-3}$ | $6.84 \times 10^{-3}$ |
| 2 | 4096 | 4 | $3 \times 10^{-4}$ | $3.54 \times 10^{-3}$ | $6.64 \times 10^{-3}$ |
| 3 | 256 | 4 | $1 \times 10^{-4}$ | $4.88 \times 10^{-3}$ | $8.67 \times 10^{-3}$ |
| 4 | 256 | 4 | $3 \times 10^{-4}$ | $4.23 \times 10^{-3}$ | $8.10 \times 10^{-3}$ |
| 5 | 256 | 8 | $3 \times 10^{-4}$ | $6.17 \times 10^{-3}$ | $1.15 \times 10^{-2}$ |

### D.1. Reconstruction performance of 3D-VQ-GAN

This experiment investigates the influence of two critical hyperparameters—codebook vocabulary size and compression rate—on the performance of the 3D-VQ-GAN model (Appendix E.2). The codebook vocabulary size specifies the number of discrete latent vectors in the codebook $\mathcal{Z}$, which serve as building blocks for representing input data in the latent space. A larger vocabulary size enables the model to capture finer details in 3D CT scans but increases computational demands. In contrast, the compression rate controls the degree of dimensionality reduction during encoding, reducing the complexity of the latent space representation. While a higher compression rate simplifies the model, it risks losing information and compromising reconstruction quality. To identify the optimal configuration, the model was trained on a training dataset and evaluated on internal and external test sets (Table 3) using quantitative metrics (e.g., reconstruction error) and qualitative visual inspection. A vocabulary size of 256 and a compression rate of 4 were chosen as they offered the best balance between detail preservation, computational efficiency, and reconstruction performance. Figure 6 showcases examples of input 3D CT scans and the corresponding reconstructions generated by VQ-GAN models trained with different hyperparameter settings.

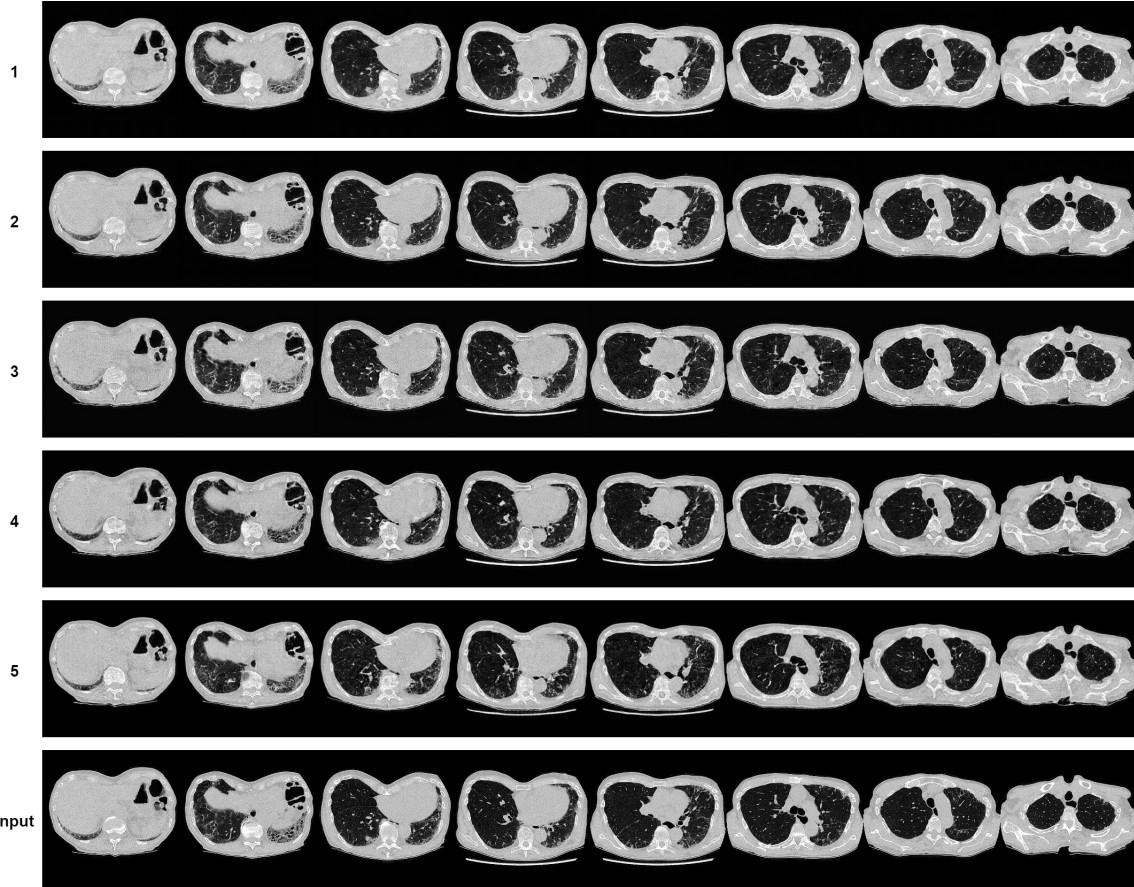

Figure 6: Examples of input 3D CT scans and corresponding CT scans reconstructed by 3D-VQ-GAN with varying hyperparameters, as detailed in Table 3

## Appendix E. Details of the Methodology

### E.1. Problem formulation

We denote the irregularly sampled longitudinal 3D imaging data of a subject as $\mathcal{X}_\mathcal{T} = \{\mathbf{X}_{t_0}, \mathbf{X}_{t_1}, ..., \mathbf{X}_{t_j}, ..., \mathbf{X}_{t_T}\}$, where each $\mathbf{X}_{t_j} \in \mathbb{R}^{D \times H \times W \times C}$ ($D$: depth, $H$: height, $W$: width, $C$: channel) and $\mathcal{T} = \{t_0, t_1, ..., t_j, ...t_T\}$ ($t_j$: the time of the $j^{\text{th}}$ observation of the subject). Each subject can have an arbitrary number of observations. In the context of IPF disease progression modelling, this translates to each patient having an arbitrary number of longitudinal volumetric lung CT scans. The corresponding mask for the region of interest is also segmented. Given $\mathcal{X}_\mathcal{T}$, the objective of this work is to build a model capable of generating synthetic 3D imaging data at any time point between $t_0$ and $t_T$, illustrating the progression of the disease within a patient over time. In this study, we use 3D volumetric CT scans of IPF patients as an example application.

The proposed method has two stages: In the first stage, a 3D-VQ-GAN is trained to reconstruct CT volumes. In the second stage, a latent (ODE) is trained to model the temporal dynamics from quantised embeddings of longitudinal CT scans generated by the encoder in the first stage, reconstructing continuous trajectories from discrete observations in the latent space (Figure 7).

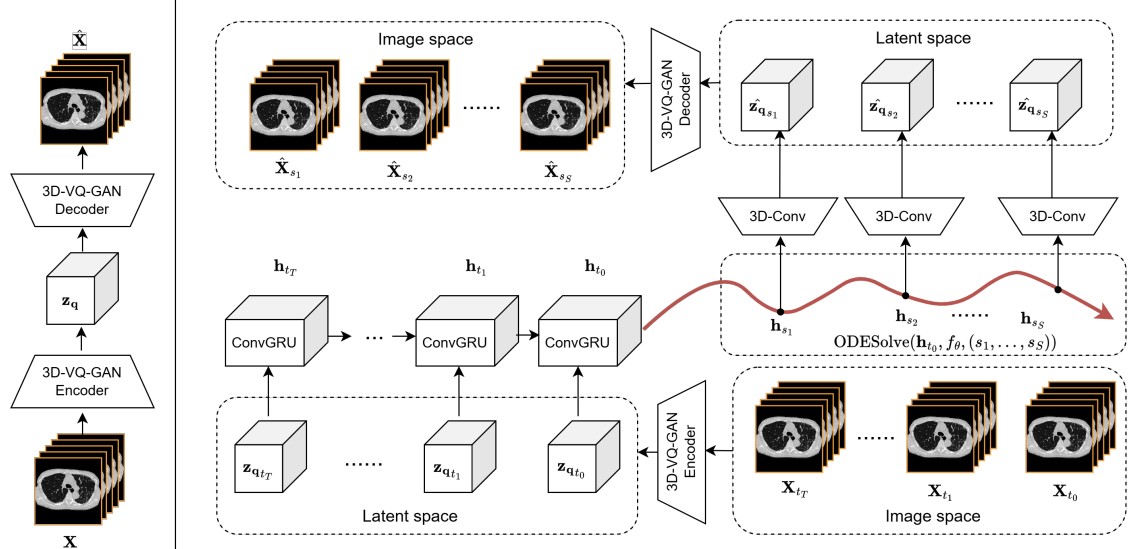

Figure 7: The overview of the proposed two-stage model. The left side is the 3D-VQ-GAN for image reconstruction. The right side is the latent disease ODE for modelling disease progression dynamics from longitudinal 3D imaging data.

### E.2. 3D-VQ-GAN for image reconstruction

To capture the dynamics of disease progression in the latent space, which can significantly decrease computational costs, the model needs to first learn an effective and compact latent

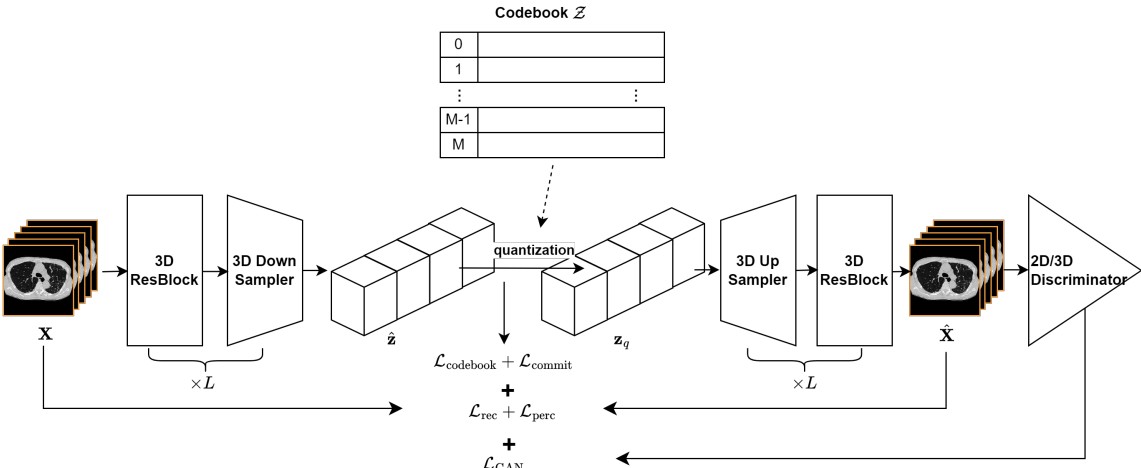

Figure 8: The overview of the 3D-VQ-GAN for image reconstruction.

representation of the input image. In the first stage, we adopt 3D-VQ-GAN (Ge et al., 2022) (Figure 8), which replaces 2D operations of the original VQ-GAN (Esser et al., 2021) with 3D operations. VQ-GAN is a variant of VQ-VAE. VQ-VAE consists of an encoder $E$ and a decoder $G$ and keeps a discrete codebook of learned representations in latent space. Given an input $\mathbf{X}$, the VQ-VAE tries to represent the input image with embeddings from the codebook in the latent space. More specifically, the encoder $E$ projects $\mathbf{X}$ into the embedding $\hat{\mathbf{z}} = E(\mathbf{X}) \in \mathbb{R}^{d \times h \times w \times c}$ in latent space followed by an element-wise quantization operation $\mathbf{q}(\cdot)$ which approximates $\hat{\mathbf{z}}$ by replacing each spatial code $\hat{\mathbf{z}}_{i,j,k} \in \mathbb{R}^c$ with its nearest neighbour in the trainable codebook $\mathcal{Z} = \{\mathbf{z}_m\}_{m=1}^M \in \mathbb{R}^c$. The discrete latent indices and embeddings after quantization are denoted as $\mathbf{c} \in \mathbb{Z}^{d \times h \times w}$ and $\mathbf{z_q} \in \mathbb{R}^{d \times h \times w \times c}$ respectively. $\mathbf{z_q}$ then goes through the decoder to reconstruct the input $\hat{\mathbf{X}} = G(\mathbf{z_q})$, using: $\mathbf{z_q} = \mathbf{q}(\hat{\mathbf{z}}) = \underset{\mathbf{z}_m \in \mathcal{Z}}{\arg\min} ||\hat{\mathbf{z}}_{i,j,k} - \mathbf{z}_m||$.

For the non-differentiable quantization operation, VQ-VAE uses a straight-through estimator (Bengio et al., 2013) which copies gradients from decoder input $\mathbf{z_q}$ to encoder output $\hat{\mathbf{z}}$ (Van Den Oord et al., 2017).

$$\mathbf{z_q} = \mathbf{q}(\hat{\mathbf{z}}) = \underset{\mathbf{z}_m \in \mathcal{Z}}{\arg\min} ||\hat{\mathbf{z}}_{i,j,k} - \mathbf{z}_m|| \tag{1}$$

The VQ-VAE loss $\mathcal{L}_{\text{vqvae}}$ consists of three terms: reconstruction loss $\mathcal{L}_{\text{rec}}$, codebook loss $\mathcal{L}_{\text{codebook}}$ and commitment loss $\mathcal{L}_{\text{commit}}$. $\mathcal{L}_{\text{rec}}$ is used for optimizing both encoder and decoder. $\mathcal{L}_{\text{codebook}}$ is used only for optimizing the codebook by pushing embeddings in the codebook to be close to the output of the encoder. $\mathcal{L}_{\text{commit}}$ is employed to enforce the encoder commits to an embedding in the codebook (Van Den Oord et al., 2017).

$$\mathcal{L}_{\text{vqvae}} = \underbrace{||\mathbf{X} - \hat{\mathbf{X}}||_1}_{\mathcal{L}_{\text{rec}}} + \underbrace{||\text{sg}[E(\mathbf{X})] - \mathbf{z_q}||_2^2}_{\mathcal{L}_{\text{codebook}}} + \underbrace{\beta||\text{sg}[\mathbf{z_q}] - E(\mathbf{X})||_2^2}_{\mathcal{L}_{\text{commit}}} \tag{2}$$

sg[·] is the stop-gradient operator here.

In addition to the VQ-VAE loss, VQ-GAN also uses GAN loss $\mathcal{L}_{\text{GAN}}$ and perceptual loss $\mathcal{L}_{\text{perc}}$ to improve the reconstruction quality as well as increase the compression rate. Similar to 3D-VQ-GAN (Ge et al., 2022), we use two discriminators $D_{2\text{d}}$ and $D_{3\text{d}}$. $D_{2\text{d}}$ is used to distinguish the real slice and the reconstructed slice of 2D plane. $D_{3\text{d}}$ is used to distinguish real 3D input $\mathbf{X}$ and reconstruction $\hat{\mathbf{X}}$ to encourage the consistency between slices:

$$\mathcal{L}_{\text{GAN}} = \log D_{2\text{d}/3\text{d}}(\mathbf{X}) + \log(1 - D_{2\text{d}/3\text{d}}(\hat{\mathbf{X}})) \tag{3}$$

Perceptual loss (Zhang et al., 2018) measures the distance of the true input and reconstruction in the feature space of a VGG network (Simonyan and Zisserman, 2014).

$$\mathcal{L}_{\text{perc}} = \sum_l w_l ||\text{VGG}^{(l)}(\hat{\mathbf{X}}) - \text{VGG}^{(l)}(\mathbf{X})||_1 \tag{4}$$

$\text{VGG}^{(l)}(\cdot)$ extracted the features of $l^{th}$ layer of VGG. $w_l$ is a learned weight for scaling. The overall loss of 3D-VQ-GAN $\mathcal{L}_{\text{3D-VQ-GAN}}$ would be

$$\begin{aligned}
&\min_{E,G,\mathcal{Z}} \left( \max_{D_{2\text{d}},D_{3\text{d}}} (\lambda_{\text{GAN}}\mathcal{L}_{\text{GAN}}) \right) + \\
&\min_{E,G,\mathcal{Z}} (\lambda_{\text{perc}}\mathcal{L}_{\text{perc}} + \lambda_{\text{rec}}\mathcal{L}_{\text{rec}} + \mathcal{L}_{\text{codebook}} + \beta\mathcal{L}_{\text{commit}})
\end{aligned} \tag{5}$$

### E.3. Disease ODE: modelling latent disease progression dynamics

#### E.3.1. Overview

As shown in Figure 7, given the trained encoder of 3D-VQ-GAN, longitudinal input 3D imaging data $\mathbf{X}_{t_0}, \mathbf{X}_{t_1}, ..., \mathbf{X}_{t_T}$ for each patient can be projected to a series of quantized embeddings $\mathbf{z}_{\mathbf{q}_{t_0}}, \mathbf{z}_{\mathbf{q}_{t_1}}, ..., \mathbf{z}_{\mathbf{q}_{t_T}}$. This sequence of embeddings can be considered as samples from the continuous disease trajectory of that subject in the embedding space. To reconstruct the continuous trajectory from discrete observations, we adapt the common latent ODE structure, an encoder-decoder-based latent-variable time series model. In this application, the primary focus lies on capturing changes within specific Regions of Interest (ROIs), i.e. lung area. To isolate and emphasize these areas in the analysis, we apply a masking technique that excludes regions outside of the ROI in the latent embedding series. The overview of the latent disease ODE is shown in Figure 7. Firstly, we use convolution-based gated recurrent unit (3D-ConvGRU) neural network (Ballas et al., 2015) as an encoder to embed the input sequence $\mathbf{z}_{\mathbf{q}_{t_0}}, \mathbf{z}_{\mathbf{q}_{t_1}}, ..., \mathbf{z}_{\mathbf{q}_{t_T}}$ into a latent initial state $\mathbf{h}_{t_0}$. Then, the continuous latent trajectory can be generated by using an ODE solver given $\mathbf{h}_{t_0}$. Finally, the latent trajectory is projected back to the embedding space of 3D-VQ-GAN to get embeddings $\hat{\mathbf{z}_{\mathbf{q}_{s_1}}}, \hat{\mathbf{z}_{\mathbf{q}_{s_2}}}, ..., \hat{\mathbf{z}_{\mathbf{q}_{s_S}}}$ at any target timesteps $\mathcal{S} = \{s_1, s_2, ..., s_S\}$. Feeding this sequence to the trained decoder $G$ of 3D-VQ-GAN can reconstruct the 3D imaging data at target timesteps $\hat{\mathbf{X}}_{s_1}, \hat{\mathbf{X}}_{s_2}, ..., \hat{\mathbf{X}}_{s_S}$.

#### E.3.2. Latent encoder: 3D-ConvGRU

ConvGRU (Ballas et al., 2015) leverages convolutions within the GRU framework, enabling it to simultaneously process both spatial and temporal information in sequential data.

Building on the concept of ConvGRU from (Ballas et al., 2015), which employs 2D convolutions, this application utilizes 3D convolutions instead. This modified unit is referred to as 3D-ConvGRU and the corresponding update function is named 3D-ConvGRUCell. Given the sequence of quantized embeddings $\mathbf{z}_{\mathbf{q}_{t_0}}, \mathbf{z}_{\mathbf{q}_{t_1}}, ..., \mathbf{z}_{\mathbf{q}_{t_T}}$, 3D-ConvGRU models the time series by making next-step prediction based on previous hidden state in an autoregressive way. The 3D-ConvGRU is run backwards as suggested by (Chen et al., 2018) and can be formulated as: $\mathbf{h}_{t_{i-1}} = \text{3D-ConvGRUCell}(\mathbf{h}_{t_i}, \mathbf{z}_{\mathbf{q}_{t_{i-1}}})$, where $\mathbf{h}_{t_i}$ is the hidden state on $t_i$.

### E.3.3. Latent decoder

The latent decoder comprises three components: a neural ODE, a 3D convolution layer, and a linear composition layer. This decoder architecture is adapted from (Park et al., 2021).

The neural ODE defines a continuous hidden state $\mathbf{h}(t)$ which is the solution of an ODE initial-value problem (IVP) as follows (Chen et al., 2018; Rubanova et al., 2019). Here the initial status is $\mathbf{h}_{t_0}$ produced by the above 3D-ConvGRU.

$$\frac{d\mathbf{h}(t)}{dt} = f_\theta(\mathbf{h}(t), t), \qquad \mathbf{h}(t_0) = \mathbf{h}_{t_0} \tag{6}$$

$f_\theta$ is a neural network parameterized by $\theta$ and $f_\theta$ defines the dynamics of $\mathbf{h}(t)$. By employing a numerical ODE solver, the hidden states $s_1, s_2, ..., s_S$ at any target timesteps can be obtained based on the initial status $\mathbf{h}_{t_0}$. This method excels by accommodating more complex dynamics within the latent state, as opposed to relying on restrictive assumptions like linearity (Sauty and Durrleman, 2022; Kim et al., 2021). This enables more flexible disease progression modelling by using neural networks to directly parameterize the changes in the hidden state. Subsequently, a single 3D convolution layer takes two hidden states $\mathbf{h}_{s_i}$, $\mathbf{h}_{s_{i-1}}$ and outputs the difference map $\mathbf{D}_{s_i}$, approximating the difference $\Delta\mathbf{z}_{\mathbf{q}_{s_i}}$ between the current $\mathbf{z}_{\mathbf{q}_{s_i}}$ and the previous $\mathbf{z}_{\mathbf{q}_{s_{i-1}}}$.

The final output of the latent decoder $\hat{\mathbf{z}_{\mathbf{q}_{s_i}}}$ is generated by combining previous generated output $\hat{\mathbf{z}_{\mathbf{q}_{s_{i-1}}}}$ with the difference map. The loss for training the latent disease ODE comprises the sum of two components. The reconstruction loss, denoted as $\mathcal{L}_{\text{recon}}$, is defined as $||\hat{\mathbf{z}_{\mathbf{q}}} - \mathbf{z}_{\mathbf{q}}||_2^2$, and the difference loss, denoted as $\mathcal{L}_{\text{diff}}$, is calculated as $||\mathbf{D}_{s_i} - \Delta\mathbf{z}_{\mathbf{q}_{s_i}}||_2^2$.

Feeding $\hat{\mathbf{z}_{\mathbf{q}_{s_i}}}$ to the trained decoder $G$ of 3D-VQ-GAN can reconstruct the corresponding 3D image $\hat{\mathbf{X}}_{s_i}$. The generative process of the decoder can be formulated as follows.

$$\begin{aligned}
\mathbf{h}_{s_1}, \mathbf{h}_{s_2}, ..., \mathbf{h}_{s_S} &= \text{ODESolve}(\mathbf{h}_{t_0}, f_\theta, (s_1, ..., s_S)), \\
\mathbf{D}_{s_i} &= \text{3DConv}(\mathbf{h}_{s_i}, \mathbf{h}_{s_{i-1}}), \\
\hat{\mathbf{z}_{\mathbf{q}_{s_i}}} &= \mathbf{D}_{s_i} + \hat{\mathbf{z}_{\mathbf{q}_{s_{i-1}}}}, \\
\hat{\mathbf{X}}_{s_i} &= G(\hat{\mathbf{z}_{\mathbf{q}_{s_i}}})
\end{aligned} \tag{7}$$

### E.4. Visualization of codebook

Every entry in the codebook corresponds to a distinctive representation or code assigned to a specific region or pattern within the input space. Utilizing the techniques outlined in (Irie et al., 2023), we visually represent each code in the learned codebook by creating a latent representation $\mathbf{z}_{\mathbf{q}}$ using only that specific code. The resulting latent representation is then

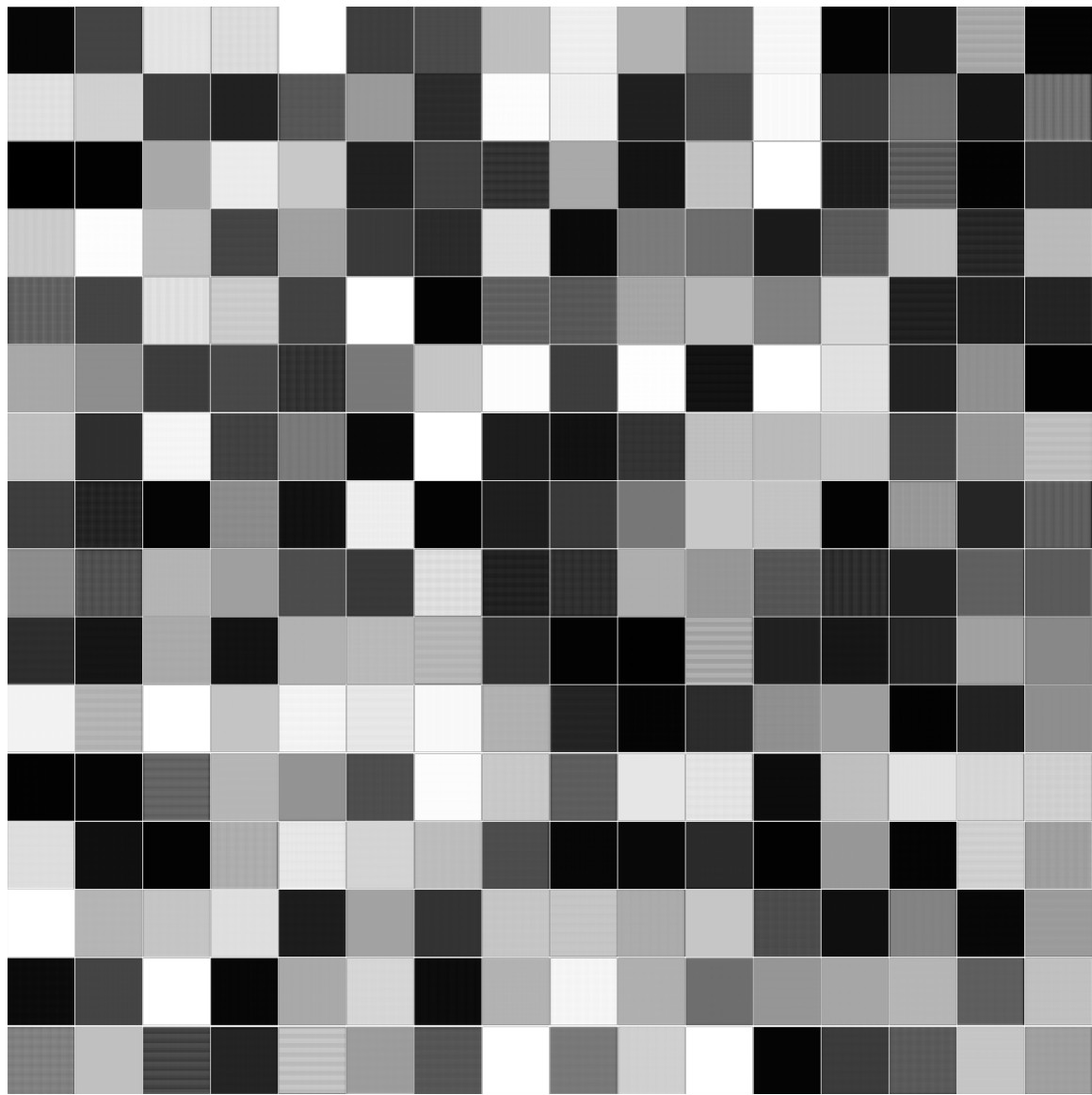

Figure 9: A visualization of the codebook ($M = 256$)

projected back into the 3D image space (see Figure 9). These visualizations exhibit varying grayscale intensities and textures, highlighting the diverse characteristics associated with different codes. This diversity within the codebook suggests that the model has effectively captured a broad array of features, enabling it to generate samples with varied and realistic qualities.

## Appendix F. More Related Work

### F.1. Synthetic medical image generation

Synthetic medical image generation proves particularly useful in various applications, classified into two types: unconditional and conditional, depending on whether constraints (e.g., images, a specific disease state, imaging modality, etc.) are applied respectively. The most common generative models used for image generation include Generative Adversarial Network (GAN) (Creswell et al., 2018), VAE (Kingma and Welling, 2013), and diffusion models (Ho et al., 2020). These models have demonstrated success in natural image generation and have shown their potential in the context of medical images. However, 3D imaging data (e.g., CT, MRI) is widely used in the medical field and generating realistic 3D images poses more significant challenges compared to 2D natural images due to the inherent complexity of three-dimensional space and the additional considerations required for realism. More specifically, unlike a 2D image with a single viewpoint, generating 3D images requires modelling the whole 3D structure which involves capturing depth information, spatial relationships, and fine details - essentially the "world behind the image". This requires not only higher computational resources but also more advanced techniques to model these detailed 3D structures with fidelity. The challenge is further amplified in the medical domain, where accurate representation of anatomical structures and simulation of physiological processes add significant layers of complexity.

Methods employing 3D-GANs have been proposed for the synthesis of 3D imaging data (Ferreira et al., 2022; Singh and Raza, 2021). However, training these models poses a considerable challenge due to increased computational and memory demands. In response to this issue, several memory-efficient 3D-GANs have been introduced (Sun et al., 2022; Uzunova et al., 2019). GAN-based models are widely used in generating volumetric medical imaging data (Ferreira et al., 2024). However, these models face additional challenges including mode collapse, non-convergence, and lack of interpretability. Mode collapse occurs when GANs fail to capture the full diversity of training data distribution and get stuck producing a limited set of outputs (Bau et al., 2019). Training GANs can also be difficult due to the need to balance and synchronize discriminator and generator. This often requires careful hyperparameter tuning and network architecture design to ensure convergence. Additionally, GANs typically lack interpretability, as it is challenging to understand what GANs have learned in the latent representation (Shen et al., 2020). In contrast, VAE (Kingma and Welling, 2013) has gained popularity for its explicit latent space representation and stable training process.

The Vector Quantized Variational Autoencoder (VQ-VAE) (Van Den Oord et al., 2017), a variant of VAE, was introduced to learn a discrete latent space, where continuous latent representations in the traditional VAE are quantized to discrete codes using a codebook. While the discrete latent space enhances efficiency and compactness, it also limits the model's ability to capture the full complexity of the input data, leading to blurry generated images. To address this limitation, VQ-VAE-2 (Razavi et al., 2019) utilized hierarchical multi-scale latent maps for large-scale image generation. VQ-GAN (Esser et al., 2021), a variant of VQ-VAE, incorporated a discriminator and perceptual loss, combining the strengths of both VQ-VAE and GAN to generate high-resolution images. Ge et al. (Ge et al., 2022) extended VQ-GAN for image modelling to 3D-VQ-GAN for video modelling.

While pure GAN-based models dominate 3D medical image generation, VAE and VQ-VAE architectures are gaining traction. Existing applications focus primarily on brain and heart MRI scans (Liu et al., 2024; Tudosiu et al., 2020). Khader et al. (Khader et al., 2023) demonstrated the potential of 3D-VQ-GAN for lung CT scans by combining it with transformers to generate realistic 3D CT scans based on a set of 2D radiographs. This highlights the capability of 3D-VQ-GAN for compressing volumetric lung CT scans.

Diffusion models (Ho et al., 2020; Croitoru et al., 2023) represent another emerging area in generative modelling. Diffusion models are a powerful class of probabilistic generative models that can learn complex distributions. These models initiate with a forward diffusion stage, where the input data is iteratively perturbed by adding noise, ultimately resulting in purely Gaussian noise. Subsequently, the models learn to reverse this diffusion process, aiming to reconstruct the original noise-free data from the noisy data samples (Kazerouni et al., 2023). While diffusion models can generate diverse and high-quality images, their application in 3D imaging data synthesis remains underexplored due to their high computational cost and low sampling efficiency compared to VAE and GAN families. Khader et al. (Khader et al., 2022) employed a diffusion model in the lower-dimensional latent space of VQ-GAN rather than the image space to reduce computational costs and increase sampling efficiency.

### F.2. Temporal synthesis

Temporal synthesis can be viewed as a challenge that involves combining static image synthesis with temporal dynamics modelling. Recurrent Neural Networks (RNNs), Long Short-Term Memory (LSTM), and Gated Recurrent Unit (GRU) are frequently employed in temporal analysis. In addition, the recent success of transformer-based models in sequential data processing has sparked considerable interest due to their potential in modelling longitudinal data (Li et al., 2022).

Previous research on temporal synthesis often concentrated on video generation, with GAN-based models being the predominant methods inspired by the success of GANs in image generation. However, GAN-based approaches may face challenges in capturing long-term dependencies. Additionally, generating high-resolution frames or long video sequences presents difficulties due to prohibitively high memory and time costs during both training and inference (Ge et al., 2022). Some methods explore non-GAN-based generative models for video generation. Models presented in (Yan et al., 2021; Ge et al., 2022; Le Moing et al., 2021) employ VQ-VAE-based models and transformers for video generation, while (Ho et al., 2022; Voleti et al., 2022) utilize the diffusion model for video generation. Computational complexity, extended inference times, and temporal consistency remain open questions for these models. Other works, such as (Kanaa et al., 2021; Park et al., 2021; Xia et al., 2022), combine a typical encoder-decoder architecture with latent neural ODEs to capture temporal dynamics in the latent space for continuous-time video generation.

Prior research in temporal synthesis for 3D imaging data has primarily focused on modelling two areas: normal brain ageing and disease progression in AD. This is often achieved by generating synthetic longitudinal brain MRIs. Normal ageing of the brain is characterized by a gradual loss of grey matter, particularly in the frontal, temporal, and parietal regions (Lorenzi et al., 2015). In contrast, the brain morphology observed in AD patients reflects

a combination of both normal ageing and pathological matter loss specific to the disease (Lorenzi et al., 2015). These two processes can be modelled independently or jointly using temporal synthesis techniques (Sivera et al., 2019). Ravi et al. (Ravi et al., 2022) introduced the 4D-Degenerative Adversarial NeuroImage Net (4D-DANI-Net), a model crafted to generate high-resolution longitudinal MRI scans that replicate subject-specific neurodegeneration within the contexts of ageing and dementia. TR-GAN (Fan et al., 2022) was conceived to predict multi-session future MRIs based on prior observations, utilizing a single generator. Sauty et al. (Sauty and Durrleman, 2022) proposed a model that amalgamates a VAE with a latent linear mixed-effect model to estimate linear individual trajectories in latent space, enabling the sampling of patients' trajectories at any given time point. While this model transforms observations at discrete time points into continuous disease progression trajectories, it relies on a strong assumption about linear trajectories in latent space. This linear assumption provides a simplified depiction of disease progression, but it falls short in capturing the inherent complexity observed in real-world disease dynamics. More flexible and adaptive approaches are needed to characterize disease progression trajectories effectively (Kim et al., 2021). To address this limitation, Martí-Juan et al. (Martí-Juan et al., 2023) employed a recurrent VAE where the latent space is parametrized with an RNN, defining more flexible disease evolution dynamics.

These temporal synthesis models for 3D imaging data offer substantial potential for clinical applications. These applications include: 1) Data Imputation: Longitudinal datasets are quite useful for the study of progressive diseases. However, longitudinal datasets often contain missing or incomplete data due to various reasons, such as missed appointments, dropout from the study, etc. (Fan et al., 2022) and (Fan et al., 2024) complement missing sessions for longitudinal MRI dataset expansion based on these models. 2) Assessing Treatment Efficacy: These models can create simulated longitudinal data that closely mimics the natural disease progression. This allows researchers to compare longitudinal imaging biomarkers between treated and untreated individuals at any point in time. By observing how the simulated disease course is altered by treatment, researchers can gain valuable insights into the treatment's effectiveness in slowing or even halting the disease process. This information can be crucial for designing future clinical trials and making informed treatment decisions. 3) Discovery of Temporal Biomarkers: Temporal synthesis models offer the ability to generate rich longitudinal imaging features. Analyzing the relationships between these features over time and how they connect to clinical outcomes can provide valuable insights. Researchers can leverage this approach to unlock the underlying mechanisms of disease progression and potentially discover novel temporal biomarkers.

