# OpenReview forum: "4D-VQ-GAN: A World Model for Synthesizing Medical Scans at Any Time Point for Personalized Disease Progression Modeling of Idiopathic Pulmonary Fibrosis"
_MIDL.io/2025/Conference — MIDL 2025 Oral_

### Official Review · Reviewer_NP2q · 2025-02-18

**Confidence:** 4
**Preliminary Rating:** 5
**Recommendation:** Oral

**Summary:**

This paper introduces a deep learning algorithm to predict and interpolate longitudinal 3D lung images. It is a difficult problem, and the presented algorithm seems to perform reasonably well for a first attempt. This work suggests that there is a good potential of such methods in clinical applications.

**Strengths:**

The paper is well written, with a clear and concise explanation of the problem, previous methods on similar problems but on brain images, and the proposed method. I did not read the method with care, but it seems to be quite complete with many details on the algorithm and further testing.

**Weaknesses:**

I did not find any major weaknesses in this work to report here. Maybe I could suggest some small improvement on figure 3. I am not an expert in analyzing these images, and I would be curious to know a bit more about what is captured by the algorithm properly, and what is not, in terms of the biological features we can observe. This could be done simply with some added arrows to point to specific areas of the lung, with a comment in the text or caption.

**Detailed Comments:**

The text is very well written I don't have any minor improvements to report.

**Justification Of The Preliminary Rating:**

In my opinion, this is a good paper. I am not an expert in these deep learning methods, so I cannot judge if for example the most appropriate ones have been used, but the explanation of the algorithm is very reasonable and the results interesting.

**Questions To Address In The Rebuttal:**

I don't have anythin specific to ask, except maybe the improvment on figure 3 if it is possible/meaningful.

**Special Issue:**

No

---

> ### Author Response · Authors · 2025-03-08
> **Response to Reviewer NP2q**
>
> We thank the reviewer for the constructive suggestions and kind words.
>
> We have now modified Figure 3 based on your suggestion with the assistance from our experienced consultant thoracic radiologist. Specifically, we added arrows to point to specific areas of the lung in the images, highlighting the relevant biological features which are correctly generated by our model. Additionally, we have updated the figure caption to provide further clarification on what is captured by the algorithm and what may not be as accurately represented. We hope these changes address your concerns and enhance the clarity of the figure.

---

### Official Review · Reviewer_2kw8 · 2025-02-20

**Confidence:** 4
**Preliminary Rating:** 4
**Recommendation:** Oral, Poster
**Final Rating:** 5

**Summary:**

The method follows a two-stage training process: first, a 3D-VQ-GAN is trained to reconstruct CT volumes, and second, a Neural Ordinary Differential Equation (ODE)-based model captures the temporal evolution of the learned quantized embeddings. The model is evaluated using two datasets: a Belgian cohort with 681 patients and a UK cohort, the latter specifically for assessing CT synthesis by the 3D-VQ-GAN.

The evaluation consists of two main components: (1) assessing CT synthesis quality for both interpolation and extrapolation, and (2) analyzing imaging biomarkers computed from synthetic CT scans to predict survival, comparing their prognostic performance against biomarkers derived from real CT scans. Additionally, an ablation study is conducted to assess the impact of key components of the proposed approach.

**Strengths:**

The paper clearly outlines the problem, proposed approach, and evaluation strategy. The methodology is detailed and well-supported by an extensive appendix. The paper honestly acknowledges its limitations, including potential challenges related to generalization and disease trajectory modeling.
The work is definitely interesting. This paper tackles longitudinal disease modeling in IPF, a field with limited research, with an innovative approach and detailed analysis.

**Weaknesses:**

The main weakness of the paper would be on the evaluation and robusness. The study relies on a single training/validation split, preventing an assessment of performance variability. A more robust approach would involve cross-validation or repeated experiments to quantify the uncertainty of reported metrics.

- CT synthesis validation remains challenging, and Table 1 does not always show a clear winner, especially for extrapolation tasks.

- Survival prediction validation is a bit weak and could be more convincing. It should be more carefully described.

**Detailed Comments:**

While the methodology is well-described (besides survival prediction), some areas would benefit from further elaboration or investigation.

For CT synthesis, the lack of statistically significant differences raises concerns about the practical reliability of the generated scans. While quantitative metrics are provided, a qualitative assessment from radiologists could also strengthen the study.

The procedure for extracting and selecting biomarkers for survival prediction is not clearly explained. It is unclear how the five biomarkers were chosen and whether feature selection stability was tested. Cross-validation should be performed to evaluate the robustness of the selected biomarkers, their stability and their predictive power. What is the rational behind the choice of 5 biomarkers. As they are selected individually, is there any correlation between the selected five?

The description of "longitudinal biomarkers" is ambiguous: "We compute the longitudinal biomarker by evaluating the change in these top five significant biomarkers over the course of one year, both for real and generated CT scans."
        What does this mean in practice?  (number of real or synthetic scans used, strategy to identify additional time points were synthetic scans are generated...).
        How exactly are the longitudinal biomarkers computed and compared.
Information on patient cohorts, survival outcomes, and covariates is insufficient. This is essential for reproducibility. All details necessary for making the survival prediction reproducible should be included in the manuscript.

**Justification Of The Final Rating:**

The revised manuscript is a significant improvement over the original submission. The descriptions of biomarker extraction and the evaluation of their robustness are now much clearer. I acknowledge that a more rigorous evaluation would require substantial computational resources beyond the timeframe of the review process. Given these improvements, I have updated my rating accordingly.

**Justification Of The Preliminary Rating:**

The work presents an interesting methodological approach; however, its clinical relevance warrants further exploration. Key areas require clarification, particularly the survival prediction process and the biomarker extraction methodology. Additionally, the study's robustness remains challenging to assess, and there may be sensitivity to sampling effects. A more thorough investigation into these aspects would enhance the study's applicability and reliability.

**Questions To Address In The Rebuttal:**

The survival prediction methodology needs more detailed explanation, especially regarding the biomarker extraction process and the techniques used to assess changes over time.
The validation of the approach should be improved by addressing statistical robustness to better assess performance variability.

---

> ### Author Response · Authors · 2025-03-08
> **Response to reviewer 2kw8**
>
> We thank the reviewer for the constructive suggestions and kind words highlighting our work as interesting, detailed, and clearly written. We now start to address the reviewer’s comments in detail. We also updated our Figure3 , and a new cross-validation based biomarker selection process (page5 and appendix B).
>
> **R2.W1. CT synthesis validation remains challenging.**
>
> The qualitative similarities between the real and generated CT scans as judged by an experienced consultant thoracic radiologist in our team provides hope that the methods outlined in this paper could mark the first step to data-driven progression estimation in respiratory disease. The quantitative results in the Table1 were only used as suggesting guidance.
>
> **R2.W2. Survival prediction validation needs more detailed explanation.**
>
> We have added an extra cross-validation based biomarkers selection method and we have updated our survival prediction results according to those more robust biomarkers. The more detailed description of our survival analysis pipeline and the new results can be found in the updated manuscript section 2 Method. Survival analysis and biomarker discovery on page 5 and Table 2 on page 9.
>
> **R2.D1. A radiologist's qualitative assessment could strengthen the study.**
>
> We have enhanced our Figure3 with more qualitative analysis from an experienced consultant thoracic radiologist in our team, the new figure 3 can be found in the new manuscript page 7.
>
> **R2.D2. The procedure for selecting biomarkers needs clarification. Cross-validation should also be performed.**
>
> We’ve updated our biomarker selection based on a cross-validation based method. As a result, while three biomarkers remained unchanged, two were replaced with more predictive ones, improving performance. We added more detailed clarification in both Section 2 (page 5) and Appendix B.
>
> **R2.D3. Biomarker extraction and selection follow a structured pipeline to ensure the identification of clinically meaningful and statistically robust imaging biomarkers.**
>
> We have updated our biomarker selection in section 2 method, survival analysis and biomarker discovery, page 5 and Appendix B. Here we provide a snapshot:
>
> - First, the trained 3D-VQGAN encoder is used to extract discrete codebook indices from CT scans. The frequency distribution of these indices is computed as a normalized histogram, where each bin represents the proportion of a specific imaging pattern in the scan. This histogram serves as a vector of candidate biomarkers.
>
> - Second, we perform survival analysis using five-fold cross-validation. The original training dataset is divided into training and validation datasets. In each fold, univariate Cox models are applied to each biomarker of baseline CT scans in the training dataset, adjusting for age, sex, and smoking status. Biomarkers are selected based on three criteria: HR > 1 (indicating that higher biomarker values are associated with increased mortality risk), p-value < 0.01 (ensuring statistical significance), and C-index > 0.5 (demonstrating predictive ability). A biomarker must satisfy these criteria in all five folds to be considered robust.
>
> - Finally, the top 5 biomarkers with the highest mean validation-set C-index are selected as final biomarkers.
>
> **R2.D4. What is the rationale for Selecting 5 Biomarkers?**
>
> We consulted our consultant radiologist who assessed our CT scans for key imaging features like honeycombing, reticulation, ground-glass opacity, traction bronchiectasis, and emphysema to evaluate disease progression in IPF. While our algorithm may uncover new patterns, we expect it to at least identify 5 biomarkers. As shown in the new Figure 3, we also pointed out the successfully generated imaging biomarkers, in page 7.
>
> **R2.D5. Is there any correlation between the selected five?**
>
> The selected five biomarkers exhibit weak to moderate correlation, with pairwise correlation coefficients all below 0.64 and an average correlation of 0.41.
>
> **R2.D6. The description of 'longitudinal biomarkers' is unclear. Information on patient cohorts, survival outcomes, and covariates should be included in the manuscript.**
>
> For each patient, we use their first two available CT scans to synthesize both their second available scan and an additional follow-up scan one year later. The synthetic second scan serves as a predicted version of the actual second scan. We then extract the five selected cross-sectional biomarkers from both real and synthetic scans at two time points: the second scan and the one-year follow-up. The longitudinal biomarker is defined as the change in these biomarkers between the two time points, separately computed for real and synthetic scans. By comparing these changes, we evaluate the consistency between real and predicted biomarker trajectories. We have revised the manuscript in Section 2 (page 5). We have added details on patient cohorts, survival outcomes, and covariates in Section 3 (page 6) of the revised manuscript.

---

> > ### Comment · Area_Chair_rUZC · 2025-03-14
> >
> > Dear Reviewer,
> >
> > does this address your concern?

---

### Official Review · Reviewer_KLxB · 2025-02-21

**Confidence:** 4
**Preliminary Rating:** 2
**Recommendation:** Poster

**Summary:**

This paper presents a method for modeling and predicting the progression of IPF. The model includes a VQ-GAN to extract representation from CT volumes and a NOTE on the latent representation to model progression. The experiments are performed on a longitudinal dataset from 219 IPF patients, on which the authors demonstrated the feasibility of the presented method in interpolating and extrpolating over time, as well as survival outcome prediction.

**Strengths:**

Longitudinal image modeling and forecasting is an under-explored area in medical image research.

The experimental results demonstrate the feasibility of the presented method to model and forecast progression in IPF.

**Weaknesses:**

The descriptions of the presented model lack detail in several key aspects (see details below).

The presented methodology is of limited novelty.

The data for the development model are not clearly described, especially regarding the longitudinal nature of the problem/model. It is not clear how many time points on average are available per subject to train the model. It is also not clear what are the IPF progression label available for the survival outcome prediction.

The experiments lack comparison with related works both in disease progression modeling and in image synthesis.

It is not clear why the feature extraction and dynamic modeling cannot be done end-to-end, nor is there ablation for this component.

**Detailed Comments:**

Descriptions of the presented model lack several critical aspects of details.

1. The skip connection component used in the NODe is counter-intuitive, since as a NODE it is already modeling the “residual” of zt at adjacent time-points (i.e., dz/dt as residual between zt and zt+1). What does adding residuals on top of the ODE solutions mean?

2. The motivations for extracting the latent representations and temporal modeling in two separate steps is not clear.

3. It is not clear what are available as the GT to evaluate the generated images in both interpolation and extrapolation tasks. It seems that the testing is the same among all patients (0 and 2 years as inputs, and 3.5 year for extrapolation (and year 1.5 for interpolation). Does that mean the availability of images is very uniform across patients? How does the model apply in settings where availability of images over time is different from patient to patient, which is a common setting in practice.

**Justification Of The Preliminary Rating:**

This paper addresses an important and under-addressed area of research. The method makes sense but is limited in novelty. The descriptions of the model include some confusing component (e.g., the relation between the residual learning and the NODE) and the descriptions of the data and training lack critical details regarding the longitudinal nature of the model. The experimental evaluation lacks comparison with prior arts.

**Questions To Address In The Rebuttal:**

1. How many time points on average per patient are used to train the model.

2. What IPF event labels are available over time for survival analysis.

3. What does the skip connection imply when NODE is already modeling the residual between adjacent time steps?

---

> ### Author Response · Authors · 2025-03-08
> **Response to Reviewer KLxB**
>
> We thank the reviewer for the constructive suggestions. We now start our responses to each comment.
>
> **R1.W1. Limited novelty.**
>
> We respectfully disagree with the reviewer, as the other reviewers regard our method and results “interesting”. Our 4D VQ-GAN is the first model for generating longitudinal HRCT lung volumes and the first to integrate a 3D VQ-GAN with an NODE for disease progression exploration. It is the first to capture the aggressive, stochastic textural patterns progression in lung scans, unlike previous methods that focused on small, smooth changes in brain MRIs, which are much smaller than lung scans and therefore computationally easier to study.
>
> **R1.W2. The data description lacks clarity.**
>
> We have now included more descriptions of the data in the revised manuscript in Section 3, page 6. Also, on average, each subject has at least two, approximately three CT scans. In our survival analysis, the primary outcome variable (“label”) is time-to-death.
>
> **R1.W3. The experiments lack comparison with related works.**
>
> To the best of our knowledge, no prior work has explored the progression of CT scans in respiratory diseases. Existing image synthesis methods focus on brain MRIs, which are unsuitable for our application, as they capture only smooth, localized changes. In contrast, fibrosis progression in CT scans involves aggressive textural changes across the lung. This motivated our model design. Additionally, training on HRCT scans is computationally expensive (10 days per run), limiting resources for adapting previous methods not designed for our application.
>
> **R1.W4. Why not end-to-end.**
>
> Separating latent representation extraction from temporal modelling provides key advantages, which we have emphasized in the revised manuscript (Section 2, page 3).
>
> - Computational efficiency: HRCT scans make fitting a high-dimensional NODE in the original space impractical. Instead, we project images into latent space for disease progression modelling and then reconstruct them into image space. Applying temporal models in the latent space rather than the original space has been validated in a recent parallel, concurrent work Genie (ICML 2024) in video game generation, and many other studies in other applications.
>
> - Training stability: Independent optimization of spatial and temporal components prevents gradient interference, leading to more stable convergence. The encoder focuses on spatial features, while the temporal model captures disease dynamics without being affected by image complexity.
> Fast inference: Our design also allows fast one-step inference for generating new scans at any time point.
>
> - Transfer learning opportunities: Pre-trained encoders can extract meaningful feature representations from large, diverse datasets before fine-tuning for temporal modelling.
>
>
> **R1.D1. What does adding residuals on top of the ODE solutions mean?**
>
> We emphasize three key benefits, which are now highlighted in the revised manuscript (Section 2, Page 4):
>
> - Improved Training Stability and Efficiency: Empirical results show that incorporating skip connections stabilizes training and accelerates convergence in our case. This was also reported in previous image synthesising work across time points (Vid-ode: Continuous-time video generation with neural ordinary differential equation, Park, et al, AAAI 2021). By learning residuals of latent embeddings, gradient propagation focuses on meaningful progressive pathological changes. In contrast, directly modelling raw latent embeddings complicates optimization by forcing the network to learn both stable anatomical structures and progressive changes. Our ablation experiments confirm that modelling residuals leads to superior performance, highlighting the robustness of this design.
>
> - Reduced Numerical Integration Error: Directly modelling differences between states minimizes error accumulation during numerical integration.
>
> - Anatomical Preservation: The skip connection ensures predictions are conditioned on individual baseline anatomy, preserving stable anatomical structures and enabling personalized disease progression modelling.
>
> **R1.D2. The motivations for the two separate steps.**
>
> Please see our previous answer.
>
> **R1.D3. What are available as the GT.**
>
> Apologies for the confusion. The number of available CT scans and their corresponding time points vary across patients. The availability of our scans is not uniform. Fig. 3 represents only one example and does not reflect the full variability in the dataset. Our data is collected from clinical practice, where imaging availability naturally differs from patient to patient. To address this, our model is designed to handle irregularly sampled time points and learns from data available for each patient. Therefore, the GT time points for interpolation and extrapolation tasks differ across patients. We’ve clarified this in the revised manuscript (Section 3, page 6).

---

> > ### Comment · Area_Chair_rUZC · 2025-03-14
> >
> > Dear Reviewer,
> >
> > does this adress your concerns?
> >
> > AC

---

### Author Rebuttal · Authors · 2025-03-08

**Rebuttal:**

Thank you for your constructive reviews and suggestions. In addition to detailed specific responses to each reviewer, we highlight our changes on the manuscript here (the updates are highlighted in red in our new manuscript, uploaded here in the supporting material):

- We have now included a more robust cross-validation based biomarker discovery process and updated our survival prediction result in Table2, page 9.

- We have included more detailed descriptions on our survival analysis validation pipeline in section 2 methods, page 5.

- We have included an updated Figure 3 on page 7, with more qualitative analysis from an experienced consultant radiologist.

**Supporting Material:**

/attachment/ee3a6155cfdb66f120af42225d3433e2bdd0addf.pdf

---

### Meta-Review · Area_Chair_rUZC · 2025-03-19

**Recommendation:** Accept (Poster)
**Confidence:** 4

**Metareview:**

This paper introduces a deep learning architecture based on GANs to predict and interpolate longitudinal 3D lung images. In the latent space, they use neural ODEs to capture temporal dynamics. Two of the reviewers were very positive about the work (strong accept), while one reviewer was more critical (weak reject). I find the temporal interpolation based on neural ODEs with 3D images sufficiently novel to warrant publication.